# Examination of Preferences for COVID-19 Vaccines in Hungary Based on Their Properties—Examining the Impact of Pandemic Awareness with a Hybrid Choice Approach

**DOI:** 10.3390/ijerph20021270

**Published:** 2023-01-10

**Authors:** Zsanett Blaga, Peter Czine, Barbara Takacs, Anna Szilagyi, Reka Szekeres, Zita Wachal, Csaba Hegedus, Gyula Buchholcz, Balazs Varga, Daniel Priksz, Mariann Bombicz, Adrienn Monika Szabo, Rita Kiss, Rudolf Gesztelyi, Dana Diana Romanescu, Zoltan Szabo, Miklos Szucs, Peter Balogh, Zoltan Szilvassy, Bela Juhasz

**Affiliations:** 1University Pharmacy, Clinical Centre, University of Debrecen, H-4032 Debrecen, Hungary; 2Department of Pharmacology and Pharmacotherapy, Faculty of Medicine, University of Debrecen, H-4032 Debrecen, Hungary; 3Institute of Statistics and Methodology, University of Debrecen, H-4032 Debrecen, Hungary; 4Department of Diabetology, Pelican Clinical Hospital, 410087 Oradea, Romania; 5Department of Medical Disciplines, Faculty of Medicine and Pharmacy, University of Oradea, 410087 Oradea, Romania; 6Department of Emergency Medicine, Faculty of Medicine, University of Debrecen, H-4032 Debrecen, Hungary; 7Department of Urology and Andrology, Clinical Centre, Kenezy Gyula Campus, University of Debrecen, H-4001 Debrecen, Hungary

**Keywords:** COVID-19, vaccine preferences, hybrid choice modeling, pandemic awareness

## Abstract

The COVID-19 pandemic has posed a huge challenge to the world in recent years. The development of vaccines that are as effective as possible and accessible to society offers a promising alternative for addressing the problems caused by this situation as soon as possible and to restore the pre-epidemic system. The present study investigated the preferences of residents in Hungary’s second-largest city (Debrecen) for the COVID-19 vaccine. To achieve this aim, a discrete choice experiment was conducted with 1011 participants, and the vaccine characteristics included in the design of the experiment were determined by qualitative methods and a pilot survey: (1) country of origin; (2) efficiency; (3) side effect; and (4) duration of protection. During the data collection at three vaccination sites, respondents were asked to choose between three vaccine alternatives and one “no choice” option in eight decision situations. Discrete choice model estimations were performed using a random parameter logit (RPL) specification with the final model extended to include a latent variable measuring pandemic awareness. The results showed that the vaccine with a Chinese country of origin is the least preferred among the respondents, while the Hungarian and the European vaccines are the most preferred. Furthermore, the increase in the vaccine efficiency level increased the respondents’ sense of utility for the vaccine; the short-term side effect was preferred to the long-term one; and the increase in the duration of protection provided by the vaccine increased the respondents’ sense of utility for the vaccine. Based on the parameter estimated for the latent variable, it can be concluded that as the level of pandemic awareness (which is more positive among people with chronic diseases and less important among health workers) increases, the choice of a vaccine option becomes more preferred among respondents compared to the “no choice“. The results of our investigation could contribute towards increasing compliance in the case of the vaccination-rejecting population, not only for COVID-19, but for any kind of vaccination procedure.

## 1. Introduction

Two and a half years after SARS-CoV-2 was declared a pandemic by the World Health Organization (WHO), several virus variants appeared and spread in many countries as they have evolved continuously, leading to genetic mutations. Circulating variants included the alpha, beta, and delta variants, and at present, the omicron variant has global public health significance. As of 16 October 2022, there were 621 million cases and more than 6.5 million reported deaths globally [1].

In Hungary, vaccination with Comirnaty started on 26 December 2020, ahead of every other country in the European Union. After both Spikevax and Vaxzevria vaccines were licensed in January, they began to be used in Hungary. From February, the Russian Sputnik V and Chinese BBIBP-CorV inactivated vaccines were also available as optional choices. The official name of the Chinese vaccine in Hungary is SARS-CoV-2 (Vero Cell) inactivated vaccine. The vaccine was given to healthcare workers first, followed by residents of nursing homes, and then other residents of the elderly and those with chronic illnesses. From March 2021 onwards vaccines have become available to all Hungarian residents.

We started our research in March 2021, and at that time, five different vaccines were available in Hungary: Comirnaty (Pfizer/BioNTech), Spikevax (Moderna/Biotech Spain), Vaxzevria (AstraZeneca), Sputnik V (Russian Gamaleja Institute) and BBIBP-CorV inactivated vaccine (Sinopharm/Beijing Institute).

Unprecedented progress has been made in the development of the above-mentioned SARS-CoV-2 vaccines: preclinical and clinical data publications started only several months after the genetics of SARS-CoV-2 became known to the general public [2]. The most clinically advanced vaccine developed by Pfizer/BioNTech, BNT162b2 (tozinameran, Comirnaty), is mRNA-based: this technique combines an engineered mRNA vaccination with lipid nanoparticles (LNPs) [3]. The genetic code for the COVID-19 full-length spike protein is provided by the mRNA, which is encapsulated in lipid nanoparticles. This approach protects the immune system from recurrent infection and avoids the hazards associated with injecting the pathogen, whether alive or attenuated, into the body [2]. The other nucleoside-modified mRNA-LNP vaccine, mRNA-1273 (elasomeran, Spikevax), encoding the SARS-CoV-2 spike protein was manufactured by Moderna in early 2020. Overall, the two types of mRNA vaccines were found to be extremely efficient in avoiding COVID-19-related hospital admissions due to the alpha, delta, and omicron variants. For all versions, vaccinated patients hospitalized with COVID-19 had considerably lower disease severity than unvaccinated patients [4]. 

The AstraZeneca’s Vaxzevria (formerly called ChAdOx1) vaccine employs a modified chimpanzee DNA adenovirus that has not been introduced to human populations and produces an immune response to the viral protein encoded in the host DNA rather than the adenovirus itself [5]. This viral vector vaccine utilizes a genetically engineered virus that cannot cause disease but encodes coronavirus proteins to stimulate an immune response safely. The vaccine elicits humoral and cellular immune responses (T-, B-, and plasma cell activation) and the development of neutralizing antibodies [3]. As regards side effects, the use of Vaxzevria was associated with increased thromboembolic risk in March 2021 [3,6]. Although a causal link between these issues and the vaccine has not been proved, it has been established that the phenomena warrant additional investigation. 

The Russian Gam-COVID-Vac, trade-named Sputnik V, is based on an adenoviral vector and was developed by the Gamaleya Research Institute of Epidemiology and Microbiology. The vaccine was created using two recombinant adenovirus vectors and came in two forms (frozen or lyophilized). According to statistics from 38 million people who have been vaccinated, Sputnik V has 97.6% efficacy against SARS-CoV-19, and 100% efficacy against severe COVID-19 illness, ranking it the most potent vaccine in the world [5]. A complete study of adverse events throughout clinical trials and mass vaccinations with the Sputnik V vaccine published by Gamaleya centre revealed that no occurrences of cerebral venous sinus thrombosis were reported [5]. 

China has led the development of classic inactivated whole-virus vaccines and has had significant success with this conventional and long-established technology. Since 17 April 2022, the inactivated vaccine BBIBP-CorV from Beijing Institute of Biological Products Co., Ltd. (Beijing, China) has been added to the WHO list of COVID-19 vaccines for emergency use [7,8]. In most of the regions, people are offered a choice from the above-mentioned vaccines, depending on their availability. Along with demographic and socioeconomic factors, personal views and reliable sources of information are crucial factors in vaccination preferences and vaccine hesitancy [9]. 

Vaccine hesitancy is a factor that impedes herd immunity, and the reasons and background features of such hesitancy may be complex. Such hesitancy is crucial in light of the COVID-19 pandemic and the potential for vaccination side effects. However, vast communities remained apprehensive about vaccination, compromising the efficacy of immunization programs, particularly in less-developed regions [10]. Although vaccination has traditionally been effective in reducing the burden of disease and mortality worldwide, public faith in vaccinations can be impacted by several issues. As a result, vaccine reluctance can cause delays and refusals, and occasionally it even causes disease outbreaks [9]. We think that people’s confidence in vaccines and even in science itself is a greatly increasing global public health issue that requires more scientific interest.

All the above technologies are similar in that none of them contains live coronavirus (SARS-CoV-2), and the major antigenic target is the surface spike protein. Hungarians can still choose which vaccine they want to receive, and the state (health insurance) covers the cost of the vaccine for Hungarian residents [11]. Vaccinations against the COVID-19 virus are carried out at established vaccination points. At significant vaccination points in Debrecen, vaccines can only be prepared and diluted by pharmacists and administered by medical doctors. 

Our research aimed to examine the preferences of the residents in Hungary’s second largest city (Debrecen) regarding the COVID-19 vaccine using a stated preference type procedure (discrete choice experiment). In addition to standard discrete choice modelling, we would like to know, through a hybrid approach, whether pandemic awareness has a significant effect in decision-making.

## 2. Materials and Methods

### 2.1. The Process of the Research

Our research was conducted from March 2021 to September 2021 during the third wave of the coronavirus pandemic in Debrecen, the second-largest city in Hungary (202, 402 people) [12]. Before the data collection, we consulted with several healthcare professionals in the field/doctors, and we also managed to conduct an online focus group interview with eight participants. The aim was to narrow down the range of factors influencing the choice of coronavirus vaccine to be tested. As a result of the process, we were able to identify seven vaccine attributes, which were as follows: (1)country of origin (USA/European Union/Hungary/Russia/China);(2)type of technology used in the production (old/new);(3)the effectiveness of the vaccine (60–70%/71–90%/more than 90%);(4)the type of possible side effect (according to the package leaflet/long-term);(5)duration of protection provided by the vaccine (6 months/12 months/lifelong);(6)the number of doses required to develop protection (1 dose/2 dose);(7)the price of the vaccine (HUF 2000/HUF 6000/HUF 10,000/HUF 14,000).

After selecting the vaccine attributes, we designed the structure of our questionnaire, which consisted of three major parts. First, our respondents had to evaluate six statements (on a scale of 1 to 5) regarding the precautionary measures recommended by the National Public Health Centre during a pandemic, and then we presented the decision situations of the discrete choice experiment that formed the basis of our research. In the last section of the questionnaire, we asked questions about the COVID-19 pandemic and collected the sociodemographic characteristics of our respondents. 

Following the design of the questionnaire, we conducted a pilot study with the participation of 83 individuals mostly working or receiving higher education, based on convenience sampling. Our goal was to provide feedback on difficult-to-understand parts of the questionnaire and to obtain preliminary information about respondents’ preferences, thus establishing a Bayesian-type experimental design. In our pilot study, we included the seven attributes presented earlier, and we created our D-efficient type design with Ngene 1.2 software [13]. The efficient type of experimental designs allow researchers to gain reliable parameter estimates with significantly lower sample size. One type of this is the D-efficient experimental design, which increases the efficiency of the design by minimizing the D-error (a determinant of the asymptotic variance-covariance matrix, which is an approximation of the real variance–covariance matrix) [14]. Our experimental design included 16 choices, each of which included three COVID-19 vaccine alternatives and one “no choice” option. Given of the high number of decision-making situations, the so-called blocking was used, so respondents were faced with only a subset of situations (eight situations). An example of the decision situation of our pilot study is shown in Table 1.

Based on the results of our pilot study, the type of technology used to make the vaccine, the number of doses required for vaccination, and the price of the vaccine were omitted from our final design (these attributes did not have a significant impact on the choices, the coefficients of these attributes do not significantly differ from zero). Using the significant coefficients, we designed a Bayesian D-efficient experimental design in which, similar to the pilot study, three vaccine alternatives and one no-choice option were included in the decision situations and blocking (we arranged the 32 decision situations of our experimental design into four blocks, so similarly to the pilot study, the respondents had to choose in only eight decision situations) to avoid the fatigue effect [15]. The vaccine attributes included in the final questionnaire are illustrated in Table 2, and an example of a decision situation is shown in Table 3.

This cross-sectional study was conducted by surveying active residents of Hajdú-Bihar County, who came to one of the three vaccination points in Debrecen (University of Debrecen, Kenézy Gyula Hospital, and the Outpatient Clinic) to receive the vaccination. Participation in the study was voluntary and the questionnaires were completed anonymously. The questionnaires were filled in at these three vaccination points in Debrecen from March 2021 to September 2021. As no distribution data are available for the studied population (vaccinated inhabitants of Hajdú-Bihar County), we cannot support the representativeness of the collected sample. It is also important to emphasize that a further limitation of our sample stems from the fact that at the time of data collection, vaccination of people under the age of 30 was already taking place in Hungary. 

### 2.2. Methodology

To examine our research aim, we used a stated preference type procedure, the discrete choice experiment (DCE). The analysis of choices in DCE is based on the theory of random utility (RUT), i.e., the decision maker assumes rational, utility-maximizing behaviour, and breaks down the total utility into an observable and a random component according to Equation (1) [16,17].
(1)$$U_{n,i,t}=V_{n,i,t}+\varepsilon_{n,i,t}$$
where *U* is the total utility, *V* is the systematic part of the utility, *ε* is the random component of the utility, *n* is the respondent, *i* is the alternative, and *t* is the decision situation.

There are several model types based on the RUT, of which the conditional logit (CL) type is widely known. It is important to note that CL is relatively easy to estimate, and its results are relatively easy to interpret, but it also has several limiting assumptions, one of the most frequently mentioned is the assumption of homogeneous preferences. This suggests that the specification assumes the same taste parameter for all decision-makers in the sample for the attributes tested [16]. In order to deal with the constraint, the so-called random parameter logit (RPL) modelling has become widespread, which allows the parameters estimated for the studied attributes to vary along a predetermined distribution among the respondents and estimates its certain parameters (e.g., mean, standard deviation) [18].

In modelling our experiment, we defined our utility function in Equation (2).
(2)$$\begin{array}{l} U_{i}\\ ={\mathrm{ASC}}_{\mathrm{No choice}}+\beta_{\mathrm{USA}}{\mathrm{Country of origin}}_{{\mathrm{USA}}_{i}}\\ +\beta_{\mathrm{European Union}}{\mathrm{Country of origin}}_{{\mathrm{European Union}}_{i}}\\ +\beta_{\mathrm{Hungary}}{\mathrm{Country of origin}}_{{\mathrm{Hungary}}_{i}}+\beta_{\mathrm{Russia}}{\mathrm{Country of origin}}_{{\mathrm{Russia}}_{i}}\\ +\beta_{60–70\%}{\mathrm{Efficiency}}_{{60–70\%}_{i}}+\beta_{71–90\%}{\mathrm{Efficiency}}_{{71–90\%}_{i}}\\ +\beta_{\mathrm{According to package leaflet}}{\mathrm{Side effect}}_{{\mathrm{According to the package leaflet}}_{i}}\\ +\beta_{6\mathrm{months}}{\mathrm{Duration of protection}}_{{6\mathrm{months}}_{i}}\\ +\beta_{12\mathrm{months}}{\mathrm{Duration of protection}}_{{12\mathrm{months}}_{i}}+\varepsilon_{i} \end{array}$$
where *ASC_No choice_* is the alternative specific constant estimated for the “no choice” option, *β* is a random parameter vector estimated for the attributes examined, and *Country of origin*, *Efficiency*, *Side effect*, and *Duration of protection* indicate the attributes examined.

To be able to take into account factors such as consumer attitudes or perceptions, which also have a significant impact on the decision, we need to use hybrid modelling. In hybrid modelling, the formula in Equation (1) can be extended with another part, a certain latent construct, according to Equation (3) [19,20].
(3)$$U_{n,i,t}=V_{n,i,t}+\lambda LV_{n}+\varepsilon_{n,i,t},$$
where *LV* is the latent variable examined and *λ* is the coefficient estimated for its effect.

When using latent variable models, we can define so-called structural and measurement equations and estimate their parameters. The former describes the examined latent construct as a function of various explanatory variables, while the latter characterizes the relationship between the latent variable and the indicators measuring it [21].

In the case of our experiment, the structural equation according to Equation (4) and the measurement equation according to Equation (5) was determined.
(4)$$LV_{n}=\gamma_{Healthcare\;worker}Healthcare\;worker_{n}+\gamma_{Chronic\;patient}Chronic\;patient_{n}+\eta_{n},$$
where *γ* denotes the parameter vector estimated for the explanatory variables examined and *η* denotes the random component of the structural equation.
(5)$$ME_{k,n}=\zeta_{k}LV_{n}+\sigma_{k,n},$$
where *k* denotes the *k*th examined indicator, $\zeta_{k}$ denotes the parameter estimated for the latent variable (for the indicator *k*), and *σ* denotes the random component of the measurement equation (in our case we modelled based on the ordered logit structure, so *l* − 1 (where *l* indicates the number of categories of the examined indicator) threshold parameters were estimated for each indicator).

In our hybrid modelling, we wanted to capture our latent variable (in our case called pandemic awareness) with six statements (respondents had to rate them on a scale of 1 to 5), which were the precautions recommended by the National Public Health Centre in a pandemic [22]:−Since the existence of the pandemic, he has avoided personal contact with friends and acquaintances and group gatherings;−Since the end of the pandemic, wash your hands thoroughly (for at least 20 s) with running water or clean your hands with an alcoholic hand sanitizer several times a day;−Since the end of the pandemic, always wear a mask and maintain a protective distance of 1.5 m as required;−Routinely cleans/disinfects frequently touched surfaces (such as tables, door handles, light switches, handles, desks, toilets, taps, sinks, and cell phones) since the pandemic has occurred;−Avoid large crowds, and crowded, confined spaces (e.g., public transportation, shopping malls) since the pandemic;−You have been keeping over-the-counter medications at home since the pandemic that may help you get through the virus (such as painkillers and antipyretics).

By raising the pandemic awareness to a latent variable model, we sought to answer whether, as the agreement with these statements increases, respondents are more likely to commit to vaccination or less prefer the “no choice” option in their choices. To answer this question, the interaction in Equation (6) was created and incorporated into the choice model.
(6)$$ASC_{No\;choice_{New\;term}}=ASC_{No\;choice}+\lambda_{No\;choice}*LV,$$
where $\lambda_{No\;choice}$ denotes the effect of the latent variable (pandemic awareness) on “no choice”.

Our hypothetical model is illustrated in Figure 1.

Our model estimates were performed with the Apollo package of the R program [23,24,25].

## 3. Results

### 3.1. Descriptive Statistics of the Sample, the Evaluation Statements Examined and the Other Issues Related to the COVID-19 Situation

Our sample contains 1011 individuals, the distributions of which are shown in Table 4.

The distributions of the statements evaluated in relation to the pandemic recommendations of the National Public Health Centre are shown in Table 5.

Based on the results of Table 5, it can be seen that in the case of the examined statements, a largely positive opinion is characteristic among the respondents. The highest agreement is with statement 3 (Since the end of the pandemic, always wear a mask and maintain a protective distance of 1.5 m as required.) and statement 2 (Since the end of the pandemic, wash your hands thoroughly (for at least 20 s) with running water or clean your hands with an alcoholic hand sanitizer several times a day.), while the greatest disagreement is with statement 6 (You have been keeping over-the-counter medications at home since the pandemic that may help you get through the virus (such as painkillers and antipyretics).) and statement 1 (Since the existence of the pandemic, he has avoided personal contact with friends and acquaintances and group gatherings).

Table 6 shows the distribution of responses to additional questions related to COVID-19.

Table 6 shows that the majority of respondents in our sample had not yet contracted COVID-19 infection at the time of data collection, but if so, its severity was considered moderate (Mean = 4.9). 19.5% of the respondents have some form of chronic illness, and nearly 17.3% of them are healthcare workers. In terms of vaccinations received, 62.2% of the respondents had already taken both doses. As for the manufacturer, it can be seen that most have received Pfizer and Sinopharm vaccines, while quite a few have been vaccinated with one of the Moderna and AstraZeneca types. Regarding the motivation for vaccination, 61.6% of respondents said they had chosen to apply for the vaccine on their own, while just under 2.8% came to this decision as a result of the media. The primary aspects of COVID-19 vaccination are the protection of one’s health and the health of one’s relatives, while a relatively low proportion have a workplace. 

The second question which regards on severity scale was clarified by the organizer for every volunteer before filling out the survey as follows:

1: Asymptomatic COVID-19 infection: Individuals who had verified positive test for SARS-CoV-2 (i.e., a nucleic acid amplification test, or an antigen test) but who had no symptoms that are consistent with COVID-19;

2–3: Mild illness: Individuals who had any of the various symptoms of COVID-19 (cough, sore throat, malaise, headache, nausea, vomiting, muscle and/or joint pain, diarrhea, loss of taste and smell) but who didn’t have a fever, and the symptoms didn’t significantly affect the daily activities/routine of the individuals;

4–5: Moderate illness: Individuals who showed any of the previously listed signs and they also had a fever and/or the symptoms significantly affected their daily activities/routine (e.g.: severe fatigue, exsiccosis due to severe diarrhoea or vomiting), but they didn’t have shortness of breath, dyspnoea, or abnormal chest imaging;

6–7: Moderately severe illness: Individuals who showed evidence of lower respiratory disease during clinical assessment or imaging, but weren’t hospitalized as they had sufficient oxygen saturation measured by pulse oximetry (SpO2 ≥ 94%);

8–9: Severe illness: Individuals who were hospitalized because of their insufficient oxygen saturation and lung function (SpO2 < 94, tachypnoea, or lung infiltrates >50%) but didn’t need mechanical ventilation;

10: Critical illness: Individuals who were hospitalized and needed mechanical ventilation and life support because they had respiratory failure, septic shock, and/or multiple organ dysfunction [26,27].

### 3.2. Discrete Choice Model Estimates in Preference Space

The results of our random parameter logit (RPL) and hybrid RPL (HRPL) model estimates based on the utility function formula according to Equation (2) are shown in Table 7. In the case of the models, a normal distribution was defined for the parameters of the examined attributes, and the estimates were performed with 1000 mlhs draws [28]. It is important to mention that, during the model estimations two respondents (whose responses were included in the results of the previous sub-chapter) were excluded due to incomplete answers.

Based on the results in Table 7, we can conclude that the no-choice option was less preferred among respondents (based on the negative and significant parameter estimation for ASC “no choice”) than the choice between hypothetical vaccine alternatives. Regarding the country of origin of the vaccine, any additional level is considered preferable to the respondents compared to the Chinese, representing the base category. The most preferred places of origin are Hungary and the European Union. Regarding the level of efficiency of the vaccine, as expected, we concluded that as it increases, the sense of the utility of respondents for vaccines increases. As for the side effect, it is considered more preferred than the long-term one according to the package leaflet. Regarding the duration of protection provided by the vaccine, it can be stated that the longer the duration of the given product is guaranteed, the greater the sense of utility towards it by the decision-makers. The results show that a significant standard deviation was estimated for each vaccine attribute examined, suggesting the presence of heterogeneity in preferences. Finally, we need to highlight the additional (λ) parameter of our model estimated in a hybrid context, for which the negative and significant parameter expresses that the “no choice” becomes less preferred as the level of the latent variable (pandemic awareness) increases among respondents (compared to the choice of a vaccine option).

### 3.3. Parameter Estimates of the Structural and Measurement Equations for the Hybrid Model

In this subsection, we will describe the estimated parameters for the structural and measurement equations related to our hybrid random parameter logit (HRPL) specification (Table 8). It is necessary to mention that we also tested several combinations of explanatory variables to construct our structural equation, and the best fit was obtained with the formula in Equation (4) (including the variables “health worker” and “chronic patient”).

From the structural equation parameter estimates (*γ* parameters) shown in Table 8, we can conclude that the level of the latent variable is significantly lower among healthcare workers (compared to non-healthcare workers) and significantly higher among chronic patients (compared to non-chronic patients).

The *ζ* parameters estimated for the measurement equations suggest that all the examined indicators have a positive and significant effect, i.e., as the level of the latent variable increases, the statements will be rated higher (increasingly agreed) by the respondents. The highest coefficients can be seen for statement 5 (Avoid large crowds, crowded, confined spaces (e.g., public transportation, shopping malls) since the pandemic) and for statement 3 (Since the end of the pandemic, always wear a mask and maintain a protective distance of 1.5 m as required.), suggesting that judgments of these statements increase the most as the level of the latent variable increases.

## 4. Discussion

The survey examined six separate statements regarding pandemic awareness. Our results showed that the participants were responsible for the guidelines and precautions necessary to protect themselves and to keep the spread of the pandemic under control. The highest agreement was with the frequent and thorough washing of hands, wearing a mask, and keeping a protective distance of 1.5 m from others, if possible. The importance and proper method of washing hands were communicated widely on television and social media; therefore, it is not surprising responders took it seriously to limit the risk of getting infected or spreading the virus. Although wearing masks and keeping proper distance in pharmacies and stores was mandated by law at the time of the survey, it seems the participants understood the importance of such precautions, as more than 87% agreed with the statement. The fact that almost 80% of the responders could avoid COVID infection for a year further corroborates that they consciously took measures to protect themselves against the virus. Besides, there may be other conceivable explanations as well to such a high avoidance rate, such as an asymptomatic course of COVID infection, or too low testing rate, not to mention the fact, that respondents were recruited from the vaccination centers, where those people go who have not yet been infected, since getting the infection means protection which makes vaccination pointless and even contraindicated. Nevertheless, awareness of the community and special policies such as the legally stipulated curfew must have contributed greatly as well. At first glance there was a considerable disagreement among respondents regarding isolation, in other words, avoiding contact with friends, relatives, and crowds, but a closer look shows that one-third of the participants were neutral about the topic, and more than twice as many respondents agreed to some extent to the statement than those who found self-isolation unacceptable (46.0% and 19.8%, respectively). Compared to the other statements, there was considerable disagreement regarding the over-the-counter medicines, such as painkillers and antipyretics, that may alleviate the symptoms of coronavirus infection. Looking at the numbers we can note that the proportion of respondents who agreed with the statement is twice as big as those who stated there is no need to keep such auxiliary medicine at home (56.8% vs. 25.1%).

The limitation of the survey may be that only people present at the vaccination points (intending to be vaccinated) filled in the questionnaires. At the time of the survey, all participants had received at least one dose of COVID vaccine or were waiting for their first vaccination. Vaccination was not mandated by the government in Hungary—except for health workers—and citizens had the opportunity to reject an offered vaccine and wait for another, thus it was possible to get vaccinated according to individual preferences. An important criterion for the study is that most of the respondents wanted to be vaccinated and did not choose a vaccine out of compulsion. According to our results in Table 6, the most popular vaccination was Comirnaty with 42.1%, then BBIBP-CorV at 32%, then Sputnik with 17.8%. This was not the main purpose of our survey, but we asked respondents which vaccine they received. Since there were considerable differences between available vaccinations in different countries, foreign reports cannot be easily compared to our survey, but it is apparent that there are distinct differences in preference between the various approved vaccines [29]. Kutasi et al. surveyed vaccine preference in Hungary and they reported 35.4%, 22.1%, and 18.5% preference of participants of Comirnaty, BBIBP-CorV, and Sputnik, respectively [10]. 

According to participants, more than 60% chose to get vaccinated on their own decision. As they were in the pandemic for more than a year by April 2021, near the end of the third wave, it is not surprising they gathered as much information as they could to build literacy about the subject so they could make their own decision. The analysis of Roy et al. reported that the most common predictors of vaccine acceptance are safety and efficacy, in other words, trust in the vaccine, but also, at least in Europe, the influence of family and friends, and social networks was significant [30]. A survey conducted by the Hungarian Central Statistical Office between December 2020 and May 2021 listed vaccine safety and the perceived seriousness of the pandemic, measured by the number of infections and COVID-related deaths as the most important factors to influence vaccination willingness [31]. In our questionnaire, 70% of the responders noted the protection of themselves or family members or acquaintances as the main reason for getting the vaccine. This is no surprise if we consider it that the participants overall expressed high pandemic awareness that they were conscious of the severity of the virus and show responsibility by taking precautions to protect themselves and others. Around 20% of responders reported vaccine passports as their main reason for taking the vaccine. One of the statements regarding pandemic awareness about avoiding personal contact with people met with mixed opinions and considerable disagreement. The vaccine passport gave much more freedom, and a lot of public events could only be attended by having one, thus it is easy to see its appeal for many to get vaccinated. Workplace ranked last as the main reason for choosing vaccination, probably also because at the time of data collection, there was no law in Hungary requiring employers to make vaccination against COVID-19 mandatory for their employees.

When in real life there is a vaccine offered, the individual might choose it even if it is not that efficient or safe, just to avoid wasting time having to wait for another appointment when the more preferred vaccine might be available. In the questionnaire’s situational decision-making, the participants could choose the most appealing vaccine without such a limiting factor. The hypothetical vaccine options were evaluated by country of origin, efficiency, side effects, and duration of protection. Regarding the country of origin, China and Russia had the lowest appeal to respondents according to our data. We must note that the Chinese BBIBP-CorV vaccine that contains inactivated virus, and the viral vector vaccine Sputnik developed and manufactured by Russia is not approved by the EU [32], which might have raised concerns in the participants of the survey. An examination conducted in Israel showed that the general population rather chose a vaccine made in the USA or UK with 60% efficacy over a Chinese or Russian vaccine with efficacy over 90%, though physicians were willing to get vaccinated by a Chinese or Russian vaccine with better efficacy if those were evaluated by FDA standards [33]. In a survey conducted in Hungary in March 2021 the vaccine-hesitant people showed generally more trust towards Chinese and Russian vaccines over those of European or American origin, but as participants have become more accepting, this tendency gradually turned around, and the portion who expressed definite will to get vaccinated, were the most confident about European and American vaccines over the Chinese and Russian ones [31]. In the survey conducted by Kutasi et al. the participants who refused a vaccine and then accepted another type, a large proportion rejected BBIBP-CorV and Sputnik for Comirnaty. At the same time, the patients rarely rejected Comirnaty or Spikevax when they were offered the first time [10]. According to the respondents the higher efficiency and longer effectiveness a vaccine has against COVID, generally the more appeal it has. Our results are corroborated by surveys from several countries [30,34,35]. In the survey, the long-term side effects were less preferred than the short-term ones. That was probably due to the fast development process of the vaccines since there was high demand for starting the vaccination widely as soon as possible, there was only limited evidence of short-term side effects, but there was no evidence about the long-term effects. People were rightfully concerned about the long-term safety of the vaccines, and if they would interact with existing conditions such as diabetes or cardiovascular diseases, and such fears can directly contribute to vaccine hesitancy [30,36,37,38].

## 5. Conclusions

Based on the results of our study, we can conclude that pandemic awareness has a significant effect on the willingness to vaccinate of Hungarian respondents. As the level of pandemic awareness increases, vaccination (choosing a vaccine option) becomes more preferred among respondents than avoiding vaccination. Nevertheless, this variable appears primarily in people suffering from chronic diseases, and among healthcare workers, it is significantly reduced to the background.

Regarding the vaccine attributes, it can be stated that the Hungarian and EU-origin vaccines are the most preferred among respondents, and the increase in the efficiency level and the duration of protection and the existence of short-term (according to the package leaflet) side effects (as opposed to longer-term side effects) also increase the respondents’ sense of utility.

The novelty of our research is manifested in the fact that, to the best of our knowledge, the impact of pandemic awareness in the context of choosing vaccination against COVID-19 has not yet been investigated through hybrid choice modeling. Furthermore, our results could contribute to increase the complience in the case of vaccination rejecting population not only in COVID-19 but any kind of vaccination procedure. Nevertheless, our research also has limitations, among which it is necessary to mention primarily the composition of the examined sample and the hypothetical choice situation resulting from the applied method. 

## Figures and Tables

**Figure 1 ijerph-20-01270-f001:**
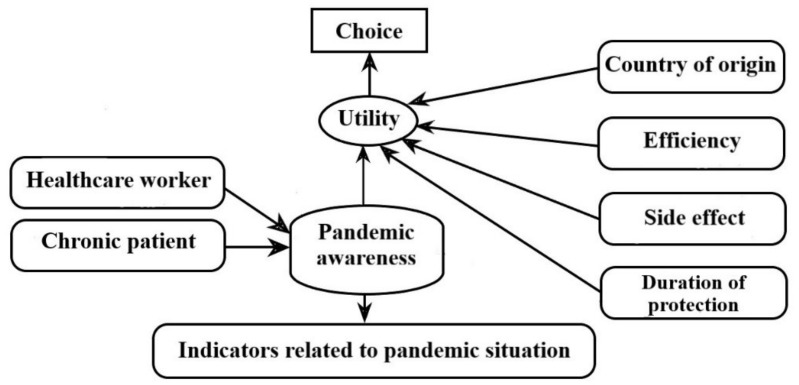
A hypothetical model of hybrid modelling.

**Table 1 ijerph-20-01270-t001:** Example of a decision situation (pilot study).

	Vaccine 1	Vaccine 2	Vaccine 3	No Choice
Country of origin	USA	China	Russia	
Type of technology	Old	New	Old	
Efficiency (%)	More than 90	60–70	71–90	
Side effect	According to the package leaflet	According to the package leaflet	Long-term	
Duration of protection	Lifelong	12 months	12 months	
Number of the dose required	2 doses	1 dose	2 doses	
Price (HUF)	6000	6000	2000	
Your choice (X):				

**Table 2 ijerph-20-01270-t002:** The attributes included in the experiment, their levels, and their descriptions.

Vaccine Attribute	Description of the Attribute	Levels of the Attribute
Country of origin	Production country of the vaccine.	USA European Union Hungary Russia China
Efficiency (%)	Efficiency level of the vaccine against the COVID-19 virus is expressed as a percentage.	60–70 71–90 More than 90
Side effect	Type of potential side effect after vaccination.	According to the package leaflet Long-term
Duration of protection	The duration of the period of protection is guaranteed by the producer after vaccination.	6 months 12 months Lifelong

**Table 3 ijerph-20-01270-t003:** An example of a decision situation.

	Vaccine 1	Vaccine 2	Vaccine 3	No Choice
Country of origin	China	European Union	USA	
Efficiency (%)	More than 90	71–90	60–70	
Side effect	Long-term	According to the package leaflet	According to the package leaflet	
Duration of protection	12 months	Lifelong	12 months	
Your choice (X):				

**Table 4 ijerph-20-01270-t004:** Presentation of the sample.

Characteristic	Sample ( *n*= 1011)	
	Count	Percentage
Gender		
Male	447	44.2
Female	564	55.8
Age category		
18–29	383	37.9
30–45	393	38.9
46–60	210	20.8
61–75	25	2.4
Highest level of education		
Primary	62	6.1
Secondary	553	54.7
Higher (minimum BSc)	396	39.2
Residence category		
Debrecen	685	67.8
Another town in Hajdú-Bihar county	221	21.8
Another township in Hajdú-Bihar county	84	8.3
Other	21	2.1

**Table 5 ijerph-20-01270-t005:** Distribution data for evaluative statements.

Statement *	1 (%)	2 (%)	1 + 2 (%)	3 (%)	4 (%)	5 (%)	4 + 5 (%)
Statement 1	8.3	11.5	19.8	34.2	26.0	20.0	46.0
Statement 2	2.4	3.5	5.9	12.7	24.1	57.3	81.4
Statement 3	0.9	2.1	3.0	9.7	28.3	59.0	87.3
Statement 4	3.4	8.9	12.3	20.8	31.3	35.6	66.9
Statement 5	3.7	7.3	11.0	22.6	27.5	38.9	66.4
Statement 6	13.4	11.7	25.1	18.1	17.9	38.9	56.8

* 1: if you strongly disagree with the statement, 5: if you completely agree with the statement. Statement 1: Since the existence of the pandemic, he has avoided personal contact with friends and acquaintances and group gatherings. Statement 2: Since the end of the pandemic, wash your hands thoroughly (for at least 20 s) with running water or clean your hands with an alcoholic hand sanitizer several times a day. Statement 3: Since the end of the pandemic, always wear a mask and maintain a protective distance of 1.5 m as required. Statement 4: Routinely cleans/disinfects frequently touched surfaces (such as tables, door handles, light switches, handles, desks, toilets, taps, sinks, and cell phones) since the pandemic has occurred. Statement 5: Avoid large crowds, and crowded, confined spaces (e.g., public transportation, shopping malls) since the pandemic. Statement 6: You have been keeping over-the-counter medications at home since the pandemic that may help you get through the virus (such as painkillers and antipyretics).

**Table 6 ijerph-20-01270-t006:** Distributions of responses for COVID-19 related questions.

Question	
Have you been infected with COVID-19? (%)	
Yes	21.4
No	78.6
If so, how severe your symptoms were? (1–10)	
Mean (standard deviation)	4.9 (2.5)
Do you have a chronic illness? (%)	
Yes	19.5
No	80.5
Are you a healthcare worker? (%)	
Yes	17.3
No	82.7
Have you already received any vaccine against COVID-19? (%)	
I got a vaccine	37.8
I received both vaccinations	62.2
If so, which vaccine did you receive? (%)	
Vaxzevria (AstraZeneca)	4.6
Spikevax (Moderna)	3.5
SARS-CoV-2/Vero Cell inactivated (Sinopharm)	32.0
Sputnik (Russian Gamaleja)	17.8
Comirnaty (Pfizer)	42.1
Why did you decide to vaccinate yourself (select only one option)? (%)	
Media/News	2.8
Healthcare professional	12.9
Family/Acquaintance	11.8
Workplace	8.1
My own decision	61.6
Other	2.8
What was the main reason for choosing COVID-19 vaccination (select only one option)? (%)	
Protecting your health	36.4
Protecting family/acquaintance	33.6
Vaccine passport (security certificate)	20.9
Workplace	6.9
Other	2.2

**Table 7 ijerph-20-01270-t007:** RPL and HRPL model estimates in preference space.

Attributes and Descriptive Data of the Model	RPL Model		HRPL Model	
	Estimates	t-Ratio	Estimates	t-Ratio
ASC (reference category: Choice of a vaccine alternative)				
No choice	−1.70 *	−21.62	−3.18 *	−12.22
Country of origin (reference category: China)				
USA	0.52 *	5.41	0.69 *	7.39
USA (standard deviation)	1.80 *	15.99	1.60 *	14.51
European Union	1.36 *	14.10	1.29 *	14.28
European Union (standard deviation)	1.92 *	19.15	1.86 *	17.76
Hungary	1.28 *	12.12	1.27 *	12.98
Hungary (standard deviation)	2.14 *	19.08	1.94 *	17.04
Russia	0.40 *	4.09	0.41 *	4.67
Russia (standard deviation)	1.75 *	15.90	1.51 *	13.56
Efficiency (reference category: More than 90%)				
60–70%	−1.48 *	−19.96	−1.47 *	−20.14
60–70% (Standard deviation)	1.24 *	13.57	1.04 *	12.13
71–90%	−0.64 *	−10.59	−0.60 *	−11.07
71–90% (Standard deviation)	0.86 *	10.25	0.50 *	4.93
Side effect (reference category: Long-term)				
In accordance with the package leaflet	1.17 *	15.11	1.08 *	15.01
In accordance with the package leaflet (Standard deviation)	1.77 *	21.98	1.69 *	20.85
Duration of protection (reference category: Lifelong)				
6 months	−2.91 *	−27.17	−2.84 *	−27.09
6 months (Standard deviation)	1.55 *	13.84	1.49 *	14.12
12 months	−1.73 *	−23.71	−1.65 *	−23.48
12 months (Standard deviation)	1.28 *	16.43	1.20 *	15.22
λ	-	-	−2.96 *	−12.31
Individuals	1009			
Observations	8072			
Parameters	19		20	
Log-likelihood (0) (for choice model)	−11,190.17		−11,190.17	
Log-likelihood (final) (for choice model)	−8301.26		−7905.16	
$\mathrm{Pseudo}\;R^{2}$	0.26		0.29	
AIC	16,640.51		31,627.91	
BIC	16,773.44		31,991.71	

Note: ASC: Alternative specific constant. λ: Effect of the latent variable for “no choice”. AIC: Akaike information criterion. BIC: Bayesian information criterion. Base levels: ASC (Choice of a vaccine alternative), China is the country of origin, Efficiency greater than 90%, Long-term side effect, Lifelong protection. * indicate statistical significance at the 1% level.

**Table 8 ijerph-20-01270-t008:** Parameter estimates of the structural and measurement equations of the HRPL model.

Structural Equation Parameters	HRPL Model	
	Estimates	t-Ratio
$\gamma_{Healthcare\;worker}$	−0.39 *	−4.22
$\gamma_{Chronic\;patient}$	0.34 *	3.44
**Measurement Equation Parameters**	**Estimates**	**t-Ratio**
$\zeta_{q1}$	0.51 *	9.97
$\zeta_{q2}$	0.51 *	8.71
$\zeta_{q3}$	0.53 *	9.00
$\zeta_{q4}$	0.40 *	8.02
$\zeta_{q5}$	0.54 *	9.95
$\zeta_{q6}$	0.34 *	6.98

Note: * indicate statistical significance at the 1% level. *γ* denotes the vector of estimated parameters for variables in the structural equation. *ζ_s_* denote the estimated parameters for the latent variable in measurement equations. Threshold parameters are shown in Table A1 in Appendix A.

## Data Availability

Data are available upon reasonable request.

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
