# Peer review of "Examination of Preferences for COVID-19 Vaccines in Hungary Based on Their Properties—Examining the Impact of Pandemic Awareness with a Hybrid Choice Approach"

_ijerph, 2023, doi:10.3390/ijerph20021270_

Round 1
Reviewer 1 Report
Dear authors,
I have received an article that looks at factors influencing COVID-19 vaccine preferences in Hungary. While this is a well-thought article, there are some concerns in this article that needs to be addressed:
Major concerns
Title & Abstract
-After reading the whole paper, the title does not reflect the entire article. I suggest incorporating "COVID-19 properties that affect preferences for COVID-19 vaccines" or some sort, as preferences in this study are mainly driven by COVID-19 vaccine properties.
- Please add pandemic awareness to the keywords
Introduction
- Factors of COVID-19 vaccine preferences and pandemic awareness are largely unexplored in the introduction. Please provide some insights.
- Lines 93-97 do not belong in the introduction
Materials and methods
- The authors need to provide when the studies are conducted (The starting point is March 2021, but until when?), what the methodology used (cohort, cross-sectional, etc.), and what is the sampling method here
- How are the participants in the pilot study recruited?
- Explain in greater detail about D-efficient type design.
- Lines 133-4 --> "Bayesian-type experimental design." What is the purpose of conducting a Bayesian-type experimental design in this study? I also did not see any baseline characteristics or pre- or post-comparisons typical of Bayesian research.
- Lines 146 --> How is the significant impact determined?
- Please move table 4 into results as well as provide the N value (not just the %)
Results
- While in some journals, it is advisable to guide readers on what this section entails, IJERPH does not require this, especially for a research article. Hence, please remove lines 258-261 and other similar sentence structures.
- Please spell out the statement again in table 5 instead of just putting "statement 1, etc." for easier reading.
- There is no explanation of "if so, how severe your symptoms were?" in the methodology section. The authors need to explain the scale, and is there any cut-off used for this Likert scale? For example, a simple cough, fever, and runny nose is a scale of 3, etc.
- Can respondents choose more than one answer for "Why did you decide to vaccinate yourlf" and "What was the main reason for choosing COVID-19 vaccination"?
Discussion
- Lines 365-366 need to be corroborated further. It seems too good to be true by one year of the pandemic, 80% of respondents avoided COVID-19 infection (especially during the third wave). What is the testing level in Hungary? Are there any special policies in place?
- Lines 275 --> Statement 6 is about using OTC medicines, not keeping them.
- Please describe the limitations of this study --> especially when this is conducted in vaccination centers. Hence only those who are willing to be vaccinated were included in this study (as evidenced by all respondents at least received one dose of vaccination)
References
- Please provide a more proper citation on references number 1, 2, 7, 9, 19, and 25
Minor concerns
- „pandemic awareness” --> It seems to be a regular use in this article; please correct them. I also suggest sending this article to a proofreading service.
- The authors use "90<"to indicate more than 90. However, it is not mathematically acceptable and pleases correct them.
- Line 58 --> What does oriental mean? Please change the vocabulary
- I believe line 190 refers to "test" and not "taste."
- Once the authors provide the abbreviation, there is no need to give the lengthened version. For example, in line 186, the authors repeated what RUT stands for when it had been explained before.
Reviewer 2 Report
The paper is well presented but I am yet to be convinced of the relevance of the scientific message to the wider community.
Reviewer 3 Report
IJERPH 17.12.2022
Review:
Examination of preferences for COVID-19 vaccine in Hungary – examining the impact of pandemic awareness with a hybrid choice approach
General feedback:
Reference section: please ensure the completeness of your references according to the stipulated journal format
Needs proofreading to make sure the language structure and grammar are adequately used. There are some minor grammar issues throughout
There is room for improvement in the paper because the literature review needs to be extended and have more research findings discussed in it
By text:
Line 56-62: please include citations/references
Line 78: missing citations – no. 4 only referring to Astra Zeneca however, in the lines before (lines 73-77), few vaccines were stated. Please include citations/references for all mentioned vaccines
Line 86: please add citation from Sinopharm itself
Line 90: … the state covers the cost of the vaccine for Hungarian residents” – please add citation/reference
Line 233: please include citation/reference
Under results:
When presented results in sentences, please put the exact figures. Please change all.
Example: Approximately 20% of respondents have some form of chronic illness – change to = …19.5% of respondents have some form of chronic illness…
Line 295-297: Please move to discussion section
“The latter is probably also due to the fact that at the time of data collection, there was no law in Hungary requiring employers to make the vaccination of COVID-19 compulsory for their employees”.
Line 303: “during the model estimations two respondents were excluded due to incomplete answers” – were these also removed from descriptive results? Please elaborate
Line 380-382: avoid repeated sentences/same info in the manuscript. Please delete to remain once
Line 387-389: those findings similar in ranking of preference with your study. Please add words to show similar in ranking of vaccine preference
Throughout the manuscript: Please change the vaccine name to actual vaccine name, not the company
(Unless if you are comparing the company to another company/ or stating company as the manufacturer/inventor)
Best wishes,
Reviewer 4 Report
Dear authors,
the manuscript could be improved.
1. For example: line 58 “ oriental vaccines 58 (Sputnik V and Sinopharm) were also available as optional choices".
Please remove the word "oriental". I do not think it is correct -Sputnic to be called an oriental vaccine. Something more it does not sound scientifically.
2. Materials and methods:
Line 117
It is written: "(3) the effectiveness of the vaccine (60 – 70%/71 – 90%/90%<);"
In my opinion this is confusing. Please make it more clear.
For example, you can change it this way: "(3) the effectiveness of the vaccine (60 – 70%; 71 – 90%; more than 90%)".
3. Table 6:
The first question "Have you gone through COVID-19 yet? (%)" does not sounds well in English. Correct it without loosing the meaning.
You can change it this way:" Have you been infected with COVID-19?" or something similar.
4. Table 6:
"Are you a healthcare worker?(%)"
Please put an interval. it should look like "Are you a healthcare worker? (%)"
5. Lines 417-418: check the grammar.
6. Why have you included in your questionnaire 2 vaccines which are not approved in EU? If somebody lives in EU he/she can not use any medicine which is not approved. And this does not depends on customers will. So you have to explain more clear why you have included the two unapproved vaccines.
In the introduction you have written "We started our research in March 2021, at this time 5 different vaccines were available 63 in Hungary: Comirnaty (Pfizer/BioNTech), Spikevax (Moderna Biotech Spain), Vaxzevria 64 (AstraZeneca), Sputnik V (Russian Gamaleja Institute) and BBIBP-CorV (Sinopharm/Bei- 65 jing Institute)."
How these two vaccines (Sputnik V and BBIBP-CorV ) are aviable in Hungary and in the same time are not approved in EU? Please explain.
Best regards!
Round 2
Reviewer 1 Report
Dear authors,
Thank you for the work done in revising this article. I have only one last concern: the introduction is now too long without adding any significant context. There is no need, in my opinion, to describe a phase III RCT of a vaccine and how it is developed in this article. Hence, I hope the authors will summarize the introduction so that it does not become a literature review.
Otherwise, everything looks great.
Reviewer 2 Report
A nice paper on how pandemic awareness affects COVID-19 vaccine acceptance.
My only concern is the authors should show more clarity on how pandemic awareness affected the choice/type of vaccine accepted by the respondents.
Reviewer 3 Report
Needs proofreading to make sure the language structure and grammar are adequately used. There are some minor grammar issues throughout
Please change the reference/citation:
https://www.elte.hu/en/content/information-on-vaccination-and-immunity-certificates-against-covid-19-virus.t.1806?m=534 = not acceptable – not original source
https://kormany.hu/hirek/ingyenes-lesz-a-vedooltas-de-nem-kotelezo = not acceptable – not original source
Throughout the manuscript: Please change the vaccine’s name to actual vaccine’s name, not the company.
(Unless if you are comparing the company to another company/ or stating company as the manufacturer/inventor); otherwise please use the vaccine’s name – it is the same if we wanted to address or compare drugs; we write the drug’s name, not the company’s name.
Reviewer 4 Report
Dear authors,
I think your manuscript sounds much better now. The changes you have done are ok.
